# The Edge of Exploration: An Edge Storage and Computing Framework for Ambient Noise Seismic Interferometry Using Internet of Things Based Sensor Networks

**DOI:** 10.3390/s22103615

**Published:** 2022-05-10

**Authors:** Frank Sepulveda, Joseph Soloman Thangraj, Jay Pulliam

**Affiliations:** Department of Geosciences, Baylor University, Waco, TX 76706, USA; joseph_thangraj@baylor.edu (J.S.T.); jay_pulliam@baylor.edu (J.P.)

**Keywords:** ambient noise seismic interferometry, Apache Cassandra, datastax enterprise, edge computing, edge storage, Internet of Things (IoT), raspberry pi, sensor networks

## Abstract

Recent technological advances have reduced the complexity and cost of developing sensor networks for remote environmental monitoring. However, the challenges of acquiring, transmitting, storing, and processing remote environmental data remain significant. The transmission of large volumes of sensor data to a centralized location (i.e., the cloud) burdens network resources, introduces latency and jitter, and can ultimately impact user experience. Edge computing has emerged as a paradigm in which substantial storage and computing resources are located at the “edge” of the network. In this paper, we present an edge storage and computing framework leveraging commercially available components organized in a tiered architecture and arranged in a hub-and-spoke topology. The framework includes a popular distributed database to support the acquisition, transmission, storage, and processing of Internet-of-Things-based sensor network data in a field setting. We present details regarding the architecture, distributed database, embedded systems, and topology used to implement an edge-based solution. Lastly, a real-world case study (i.e., seismic) is presented that leverages the edge storage and computing framework to acquire, transmit, store, and process millions of samples of data per hour.

## 1. Introduction

The availability of inexpensive low power microcontrollers, sensors, and transceivers in the late 1990s resulted in a flurry of wireless sensor network (WSN) activity in the early 2000s [1]. By the mid-2000s, WSN experts (i.e., computer and software engineers) were collaborating with geoscientists to deploy WSNs for environmental monitoring [2,3,4]. The increased spatial and temporal measurements provided by WSNs were beneficial. However, the design, development, and deployment of WSNs continued to rely heavily on WSN experts [5]. Commercially available WSN components required considerable modification before they could be deployed in real-world environments [2] and they often experienced reliability problems requiring multiple iterations of design and development before a reliable solution could be delivered [5]. WSN development was further complicated by interoperability problems resulting from the wide variety of available proprietary and nonproprietary solutions (e.g., hardware, protocols, etc.) [6].

WSNs and the Internet of Things (IoT) both originated in the late 1990s. The IoT represents a convergence of technologies allowing things (e.g., devices, objects, etc.) to communicate via the Internet [7]. Initially, interest in WSNs outpaced the IoT. Google Trend (https://trends.google.com/ accessed on 11 March 2022) data indicated year-to-year (i.e., 2004 to 2012) interest in WSNs varied between 1.3 and 4.7 times greater than the IoT. In 2013, interest in the IoT overtook WSNs and has increased year-to-year (i.e., 2013 to 2019) monotonically from 1.4 to 14.7 times greater than WSNs. Greengard [8] credits the 2007 release of the iPhone and 2010 release of the iPad with increased interest in the IoT. However, we believe the availability of inexpensive and easy-to-use embedded systems with Internet connectivity (e.g., Arduino (https://www.arduino.cc/ accessed on 11 March 2022), Gumstix (https://www.gumstix.com/ accessed on 11 March 2022), etc.) in the late 2000s contributed to increased IoT development activity.

Internet protocol (IP) is the principal network layer protocol of the Internet that provides communication among disparate networks [9]. IP was not initially considered suitable for WSNs given the limited computational resources and constrained power budget typical of WSN nodes [10]. Nonetheless, by 2010, researchers had demonstrated that IP-based WSN applications were feasible [9] and had cataloged numerous examples of IP-based WSNs within industry and the scientific community [10]. IP-based sensor networks (i.e., wired and wireless) aligned closely with the IoT. Thus, resulting in IoT-based sensor networks that reduced overall complexity, promoted interoperability, and increased scalability [11,12,13].

Currently, WSNs are contributing greatly to the IoT by transforming agriculture, healthcare, and industrial automation [14]. WSNs are considered a basic component of the IoT and the primary means of communication between machines and the future Internet [15]. The continued integration of WSNs and the IoT is expected to result in a significant increase in the number of sensors connected to the Internet [16]. 20 to 30 billion IoT devices are expected to be connected to the Internet by 2020 [17] and countless numbers of sensors will be connected to those devices. The scale and complexity of IoT data, specifically sensor network data, will be unprecedented.

Given the modest resources of IoT devices, IoT data are typically offloaded to the cloud for storage and subsequent processing [18]. Cloud computing (i.e., the cloud) consists of centralized applications offered as services via the Internet and the resources in the cloud provider’s data center providing those services [19]. The cloud, with its virtually limitless resources, supports the management of IoT devices as well as the applications and services exploiting IoT data [20]. However, cloud computing may not be the ideal solution for IoT applications where edge devices (e.g., IoT, mobile, etc.) are major producers of data [21,22,23]. The transmission of large volumes of edge device data to the cloud burdens network resources, introduces latency and jitter, and ultimately impacts user experience [24]. Moreover, excessive backhaul network traffic to the cloud negatively impacts the performance and survivability of edge devices by increasing power consumption, introducing a single-point-of-failure, and wasting edge device computing resources [21,22,23,24].

Edge computing represents an emerging paradigm where substantial storage and computing resources are placed at the edge of the network [24,25]. The “edge” is the local network typically one-hop away from an edge device [25]. The adage “compute is cheap, storage is cheaper, but data movement is very expensive” [26] and the fact that edge device performance enhancements have outpaced network performance enhancements [21] illustrate the motivation to move storage and computing resources to the edge. Edge computing allows for the better control of data (e.g., privacy, security, etc.), enhanced application performance (e.g., jitter, latency, etc.), increased scalability (e.g., data aggregation, preprocessing, etc.) and improved survivability (e.g., connectivity, reduced power consumption, etc.) [24].

Within the current IoT landscape, edge computing is considered a critical computing paradigm [25]. Edge computing is particularly useful to IoT applications where: (1) low latency is required; (2) connectivity is constrained (i.e., network capacity) or nonexistent; or (3) dense acquisition of relatively high sample rate data is occurring [18,25]. IoT applications utilizing IoT-based sensor networks to perform remote environmental monitoring (i.e., seismic) typically acquire relatively high sample rate data (i.e., 10 s or 100 s of samples per second, sps), from one, tens, hundreds, or even thousands of sensors, in locations where connectivity is either constrained or nonexistent.

As geoscientists, we intend to mitigate performance limiters commonly encountered when deploying IoT-based sensor networks for seismic monitoring by utilizing edge computing to collectively process data in a field setting without disrupting acquisition, and regularly assess the quality of our deployment strategy and results. Our goals are to reduce the cost and risk typically associated with seismic monitoring. One geophysical application that is well-suited for IoT-based sensor networks is ambient noise seismic interferometry (ANSI), in which low-amplitude body waves or surface waves traveling between sensors are extracted from large volumes of continuously-recorded ground motion data. The waves thus extracted can be used to create models of the subsurface via, for example, tomography. Ambient noise seismic interferometry offers a low-cost alternative to traditional seismic exploration methods, in which expensive seismic sources such as dynamite or Vibroseis are used. In a typical ANSI example, a potentially large set of sensors is deployed to record continuously for days or weeks and the time series recorded by a given station is cross-correlated with that of every other station in a series of time windows. Results for each time window are “stacked” (summed) to increase the signal-to-noise ratio and produce a “virtual source gather”, which is an estimate of the Green’s function for subsurface structure beneath the sensor array [27,28,29,30,31,32,33]. Recent deployments of sensor networks have comprised thousands of stations in remote areas with limited access to the Internet [31,34]. Such deployments could benefit greatly from IoT-enabled wireless sensor networks capable of processing data in the field due to the fact that, while traditional seismic techniques need only record a specified time window following a known seismic source, ANSI results require the processing and stacking of many hours (or days or weeks) of data to reveal low-amplitude waves. The quality of outcomes is less certain for ANSI unless processing can be performed and assessed in the field, before the sensors are recovered and the team has left the site.

In this paper, we present an edge storage and computing framework for IoT-based sensor networks. The framework uses common embedded systems (i.e., Raspberry Pi (https://www.raspberrypi.org/ accessed on 11 March 2022) and Tinker Board (https://www.asus.com/us/Single-Board-Computer/Tinker-Board/ accessed on 11 March 2022)) and IP-based networks to orchestrate general purpose, edge-based, computing services using a popular distributed database (i.e., Apache Cassandra). Our goal was to utilize this framework to automate the acquisition, transmission, storage, and processing of seismic data, in a field setting. The main contributions of this paper are an architecture and topology supporting IoT-based sensor network edge storage and computing that does not require a connection to the internet for continuous monitoring yet scales efficiently to the large numbers of nodes (i.e., thousands), spread over hundreds or thousands of meters, typically used in modern seismic surveys. We further provide: (1) details of the selection and review of a distributed database that complements the architecture and topology (which proved critical to the sensor network’s success); (2) recommendations regarding embedded systems to support the acquisition, storage, and processing of sensor data; and (3) a case study that validates the system in real-world remote environmental (seismic) monitoring. In this field setting, approximately 13 million samples were acquired, transmitted, stored, and processed hourly and greater than 99% of the data acquired by the edge devices (i.e., seismic stations) overall was stored, queried, and extracted for seismic processing.

## 2. Motivation

There are myriad examples in which WSNs have been proposed to replace cable-based connectivity for seismic monitoring applications [35,36,37]. However, these efforts largely focus on wireless technology itself (e.g., protocols, specification, etc.). In 2018, Jamali-Rad and Campman [38] proposed a wireless sensing framework, that utilized low-power wide-area network (LPWAN), to prioritize: (1) inherently IoT-compatible, low power, and long range wireless sensors; (2) scalable advanced wireless networking protocols; and (3) cloud storage and computing. The static context header compression initiative (SCHC) for IoT interoperability further strengthens the viability of LPWAN of IoT applications [39]. SCHC is a novel compression and fragmentation scheme for transmitting IPv6/UDP packets over LPWANs [40].

The wireless sensing framework proposed by Jamali-Rad and Campman [38] and Jamali-Rad et al. [41] relied upon a cloud paradigm (i.e., a centralized model) for remote data storage and analysis. This centralized model required that acceptable latency, data transmission rates, and data generation rates were considered when identifying applicable scenarios of interest [38,41]. Jamali-Rad and Campman [38] identified four scenarios of interest (i.e., triggered and/or continuous monitoring): (1) ground motion monitoring; (2) ambient-noise seismic interferometry; (3) microseismic fracture monitoring; and (4) quality control for active land seismic surveys. However, continuous monitoring applications required that an appropriate wireless network was available [38].

Valero et al. [42] and Clemente et al. [43] propose an in situ signal processing approach that leverages IoT technologies to develop a real-time system for performing seismic analytics within the sensor network. This approach is ideal for scenarios in which a centralized model is untenable due to constrained or nonexistent backhaul connectivity [42]. Valero et al. [42] and Clemente et al. [43] leverage their respective solutions to successfully perform autonomous, in situ, seismic imaging for thirteen nodes located a few meters apart and six nodes located approximately 15 m apart, respectively. Valero et al. [42] and Clemente et al. [43] both use MySQL (https://www.mysql.com/ accessed on 11 March 2022) to store data on individual sensor network nodes.

The challenges of acquiring, transmitting, storing, and processing seismic data are non-trivial. The seismic methods used extensively in the oil and gas industry are costly and time consuming; seismic surveys require operators to assume substantial cost and risk [44]. Likewise, seismic methods employed within the scientific community are typically costly and time consuming. The outlay costs for a single transportable array broadband seismic station (i.e., USArray (http://www.usarray.org/ accessed on 11 March 2022)) was between $30,000 to $50,000 (USD) [45]. The utilization of edge storage and computing to reduce the cost and mitigate the risk typically associated with seismic methods could have a profound impact on the oil and gas industry and the scientific community. The edge storage and computing framework described below could also prove to be particularly beneficial to the emerging Industrial Internet of Things (IIoT) or other sensor-heavy IoT applications.

Ongoing information and communication technology (ICT) development has resulted in the availability of increased computing resources and widespread connectivity enabling scientists and engineers to streamline research and create practical solutions to real-world problems [46]. In 2016, we integrated a commercially available geoscience-related digitizer (i.e., REF TEK 130-01) with an inexpensive and easy-to-use embedded system (i.e., Raspberry Pi) to provide the Raspberry Pi Enhanced REF TEK (RaPiER) platform [47]. The RaPiER proved to be an effective single-node edge-based solution. However, more complex analysis requires data from multiple nodes to be processed collectively. We built upon our previous effort and utilized easy-to-use and well-established (i.e., within the ICT community) components to develop a novel edge-based solution capable of scaling to hundreds of nodes deployed over thousands of meters.

In our study, we use an array of RaPiERs to carry out ambient noise seismic interferometry (ANSI). ANSI requires repeated cross-correlation between time series recorded by every pair of stations, computed over time and summed together to extract waves traveling between stations. RaPiERs carry out these computations while stations are still deployed in the field and can also perform real-time assessments and cataloging data characteristics as they vary over time. Results of assessments, if they are available in quasi-real-time, allow us to reconfigure the seismic deployment’s geometry or acquisition parameters, if necessary, to ensure a study’s success. Keeping track of data characteristics can help us address the important questions “How long does it take to extract seismic arrivals?” and “Have the virtual source gathers computed so far converged to an accurate estimate of the subsurface Green’s functions?” Additional processing schemes, including selectively stacking the “best” data windows, can be devised and implemented with RaPiERs (and other WSNs), as well [31]. By enabling real-time computing of seismic data at the edge, RaPiERs allow us to optimize data acquisition by maximizing the quality of results while also minimizing effort and cost in ANSI studies.

## 3. Framework Overview

### 3.1. Background

We planned to acquire data from approximately 150 seismic stations (i.e., digitizers and sensors) spaced evenly along a line slightly more than two kilometers in length. Each seismic station would acquire 250 sps data from three channels (i.e., a tri-axis geophone). However, only one channel (i.e., the vertical), downsampled to 50 sps, would be processed. It would therefore be necessary to acquire, transmit, store, and collectively process approximately 650 million data samples per day, in a field setting. At 24 bits per sample, this would result in approximately 1.8 gigabytes (GB) of data generated per day. The seismic stations would be deployed in a remote environment, without permanent support infrastructure (e.g., communication, power, etc.), for approximately one week. We intended to utilize commercially available communications infrastructure, digitizers, a distributed database, and embedded systems to minimize the cost and complexity of implementing the edge-based solution described here.

Given the requirements described above, we developed an edge-based solution relying upon an IoT-based sensor network to accomplish our goals as geoscientists. Over the course of multiple deployments, we developed a tiered architecture of embedded systems, arranged in a hub-and-spoke topology, hosting a distributed database allowing for the acquisition, transmission, storage, and hourly processing of seismic data. Thus, allowing for the adjustment, if necessary, of the configuration (e.g., acquisition parameters, geometry, etc.) and modification (i.e., the shortening or lengthening) of the duration of our deployment with high levels of confidence our goals had been achieved. Details regarding our deployment will be provided in the ‘Case Study’ section of this paper.

The collective processing of sensor network data, at the edge of the network, reduces the cost of individual sensor nodes, increases fault tolerance, and promotes flexible configuration and management of shared sensor network resources (i.e., communication, storage, and computational). However, these resources must be capable of handling the velocity, volume, etc. of sensor network data [48]. It is necessary to implement an edge-based solution where the design and arrangement of communication, storage, and computational resources support the processing of sensor network data and mitigate the inevitable network connectivity problems commonly encountered during remote environmental monitoring. The following subsections present information regarding the selection of an appropriate architecture, topology, distributed database, embedded systems, and communication infrastructure to support the edge storage and computing framework.

### 3.2. Architecture

#### 3.2.1. Background

In an effort to maximize energy efficiency, sensor networks in the early 2000s adopted an architectural design that assumed it would be necessary to store and process data, as close to the data source as possible, on nodes with modest resources [49]. This architectural design featured an application-specific and data-centric approach where the sensor networks were customized for specific applications and data were decoupled from the sensors (i.e., nodes) producing it [50]. Essentially, an egalitarian collection of sensor nodes, located within an immediate vicinity of each other, coordinate to achieve high-level objectives [50].

#### 3.2.2. Tenet Principle

Although this approach was widely adopted, Gnawali et al. [49] believed it increased system complexity and decreased manageability. Gnawali et al. [49] expected future large-scale sensor networks would be tiered (i.e., lower and upper). The lower tier would consist of many constrained sensor nodes and the upper tier would consist of fewer less-constrained nodes [49]. The upper tier reduced complexity and increased manageability via the restriction of multi-node storage and processing to the upper tier (i.e., the Tenet principle) [49]. The restriction of multi-node storage and processing to the upper tier could introduce a single-point-of-failure or be less energy efficient [50]. Nonetheless, the Tenet architectural principle complements our desire to minimize the complexity of integrating commercially available components into an overarching edge storage and computing framework.

#### 3.2.3. Proposed Architecture

Reference architectures (RA), such as INTEL-SAP RA, Edge Computing RA 2.0, etc., were developed to establish standards regarding the design of edge computing architectures and their integration with ICT [51]. Edge computing RA are typically based upon a three-layer model including cloud services as the upper layer [51]. Figure 1 illustrates a generic edge computing reference architecture.

We utilized general purpose embedded systems to implement an architecture consisting of three, Tenet architectural principle inspired, tiers (i.e., lower, middle, and upper). However, our tiers (i.e., layers) are defined by workload. The complexity of the workload and, in turn, embedded systems (i.e., hardware and software) increases from the lower to upper layers. The lower layer is responsible for the “sensing workload”, the middle layer maintains the “transactional workload” and the upper layer supports an “analytic workload”. See Figure 2 for our edge storage and computing architecture. The sensing workload consists of edge devices (i.e., digitizers and sensors) and edge gateways (i.e., lower layer embedded systems) responsible for the acquisition of raw sensor data, the pre-processing of sensor data, and its subsequent insertion into middle layer edge nodes. Middle layer edge nodes form a distributed database that stores sensor data from multiple edge devices and replicates the data to the upper layer edge nodes. The upper layer edge nodes form a distributed database that stores sensor data from multiple middle edge nodes. Sensor data within the upper layer edge nodes can be queried and extracted locally for analysis or it can be replicated to other locations (e.g., cloud, edge, etc.).

### 3.3. Topology

#### 3.3.1. Background

The distance sensor network data traverses (i.e., wired and wireless) varies from a few meters to thousands of kilometers. Network delay, errors, etc. could negatively impact the quality and timeliness of sensor network performance [48]. Considering the inevitable network connectivity problems commonly encountered during remote environmental monitoring, it is necessary to arrange communication, storage, and computational resources in a manner ameliorating the negative effects of data delay, loss, etc. in the collective processing of sensor network data [48].

#### 3.3.2. Hub-and-Spoke

In communications networks, the hub-and-spoke topology consists of nodes (i.e., spokes) connected to centralized hubs acting as switching points for network traffic [52]. Hubs are interconnected with other hubs via backbone (i.e., backhaul) networks typically carrying larger volumes of network traffic compared to hub-to-spoke network connections [52]. The hub-and-spoke topology is commonly used for computer, military, and telecommunication applications [53]. Sensor networks often adopt a hub-and-spoke topology to improve system performance by efficiently routing traffic between specific sources and destinations [53].

We adopted a hub-and-spoke topology, consisting of wired and wireless networks, to facilitate the concentration of sensor network data from the lower to upper layers of our edge storage and computing architecture. Figure 3 illustrates the specific hub-and-spoke network (i.e., a tree/star network) used. In the tree/star network, nodes are connected to a hub (i.e., a concentrator) that is, in turn, connected to a central location or other to another intermediary concentrator, in a hierarchical structure [52]. Our choice of the tree/star network was influenced by the following three factors: (1) the hierarchical structure allows for the use of concentrators with greater capability (e.g., memory, storage, etc.) as they progress upward in the tree [52]; (2) the limits (e.g., network capacity, storage and processing capabilities, etc.) of concentrators can be overcome by adding additional concentrators and redistributing nodes accordingly; and (3) the tree/star networks allows for the continued addition of nodes (i.e., scaling), provided sufficient backhaul network capacity.

### 3.4. Distributred Database

#### 3.4.1. Background

A structured collection of data, relating to some modeled real-world phenomena, is known as a database [54]. If the database structure (i.e., model) takes the form tables, it is known as a relational database [54]. The relational model has been used to develop most conventional distributed database technology [54]. A collection of multiple, logically interrelated, databases distributed over a network is known as a distributed database [54]. Distributed database management system (DBMS) is the software used to obfuscate the complexity of distributed data storage and allow for the management of the distributed database [54]. Like DBMS, a relational database management system (RDBMS) affords similar functionality to users. Microsoft Access, MySQL, and Oracle are examples of RDBMS with which readers may be familiar.

As data volumes increase, RDBMS administrators have two available scaling options: (1) the distribution of data across more machines (i.e., horizontal scaling) or (2) increasing the system performance of the existing machine (i.e., vertical scaling) [55]. Vertical scaling is simple to implement. However, it may not be the most effective scaling method given cost and technology limitations. Horizontal scaling uses relatively inexpensive commodity hardware to distribute the database across multiple systems, thus reducing the overall workload of individual systems. Unfortunately, a distributed RDBMS results in distributed transactions. This requires the implementation of a two-phase commit to prevent new transactions from executing until the prior transaction is complete and a commit response has been returned to the transaction manager [55]. As the number of transactions (i.e., data velocity) and duration of transaction processing time (i.e., data volume) increase, the RDBMS will likely encounter performance problems resulting from the way RDBMS inherently operate [55,56].

In 2016, we conducted a literature review to identify RDBMS (i.e., SQL) alternatives ideally suited for remote environmental monitoring applications [57,58,59,60,61]. Given our need to store and process 100 s of millions of samples per day, a “Not only SQL” (i.e., NoSQL) database, specifically Apache Cassandra, emerged as our database of choice. Initially created by Facebook to solve their Inbox Search problem, Cassandra leveraged Amazon’s Dynamo and Google’s Bigtable to meet challenging write-heavy (i.e., billions per day), geographically distributed, reliability, and scalability requirements [62]. Cassandra, accepted as an Apache Software Foundation (ASF) top level project in February 2010, is an open source, distributed, decentralized, multi-location (e.g., cloud, on-premises, etc.), operationally simple, nearly linearly scalable (i.e., horizontally scalable), highly available, fault-tolerant, wide-column database [55,63].

#### 3.4.2. CAP Theorem

To better illustrate the differences between SQL and NOSQL (i.e., Cassandra) we will elaborate on the consistency, availability, and partition tolerance (CAP) theorem [55]. In 2000, Eric Brewer conceived that there are three, mutually dependent, requirements present within large-scale distributed systems: consistency, availability, and partition tolerance [55]. Consistency means each node in the system returns the “correct” response, availability necessitates each request eventually receives a response, and partition tolerance requires the distributed system continue to function even when faulty connectivity has partitioned the network [64]. CAP theorem—sometimes referred to as Brewer’s theorem—states it is only possible to strongly support two of the three requirements at a time [55]. The CAP theorem was formally proved to be true by Gilbert and Lynch [65]. Figure 4 was inspired by a graphic presented by Carpenter and Hewitt [55] illustrating where a variety of datastores align along the CAP continuum. Relational databases (e.g., MySQL, SQL Server, etc.) prioritize availability and consistency and Cassandra prioritizes availability and partition tolerance [55].

In 2012, Brewer provided an updated perspective maintaining that CAP theorem’s “2 of 3” is misleading since: (1) partitions are uncommon; (2) low level choices between availability and consistency occur often; and (3) availability, consistency, and partition tolerance are continuous rather than binary [66]. Brewer’s update is germane to enterprise-grade solutions including robust network infrastructure, servers, etc. However, we believe edge-based solutions running on extremely modest hardware, regularly encountering network connectivity problems, require a database with architectural pillars (i.e., mechanisms) supporting a bottom-up approach to partition tolerance. Moreover, edge-based solutions may benefit from a more nuanced approach to partitioning (e.g., data, operational, etc.) and tunable consistency that may significantly improve the solution’s robustness to major network connectivity problems without immediately compromising availability [64].

#### 3.4.3. Apache Cassandra

##### Physical Architecture

According to Carpenter and Hewitt [55], a collection of Cassandra nodes managing a dataset are known as a cluster. A Cassandra cluster is composed of nodes (i.e., a single instance of Cassandra running on a computer, and one or more data centers); a Cassandra data center (DC) is a logical set of nodes, connected via a reliable network, that are relatively close to each other [55]. Figure 5 illustrates a Cassandra cluster consisting of two DCs, each with four Cassandra nodes. Cassandra clusters can consist of multiple DCs, often geographically distributed, containing one or more Cassandra nodes [55]. However, a minimum of four Cassandra nodes are typically required to realize the advantages of Cassandra as a distributed database. Refer to Carpenter and Hewitt [55] for additional information regarding the physical architecture of Cassandra.

##### Ring

The data managed by a Cassandra cluster are known as a ring; each node comprising the ring is assigned a range of data known as its token range [55] (see Figure 6). An individual token within a Cassandra node’s token range is identified by a 64-bit integer that represents a partition within the ring [55]. A Cassandra cluster’s tokens therefore span the range −2^63^ to 2^63^ − 1 [55]. When data are written to Cassandra, a hashing function (i.e., a partitioner) determines the data’s token value based upon its partition key [55]. The data’s token value is compared with Cassandra nodes’ token ranges, its owner-node is identified, and the data are written to the appropriate partition [55]. Cassandra is able to write data to disk quickly since its design does not require disk reads or seeks [55]. Essentially, Cassandra writes data to, likewise reads data from, disks sequentially according to the data’s partition key. This design is particularly advantageous when working with time series data that has been partitioned (i.e., bucketed) according to anticipated access patterns (e.g., hourly, daily, etc.). For detailed information regarding Casandra’s ring or write path refer to Carpenter and Hewitt [55].

##### Replication and Consistency

In Cassandra, the database object controlling the replication of data to one or more nodes or DCs within a Cassandra cluster is known as the keyspace; the user defined parameter (i.e., replication factor) determining how data are replicated across Cassandra nodes and DCs is specified in the keyspace [55]. Read queries or write operations in Cassandra include a user defined consistency level specifying how many nodes must respond before a read or write is considered successfully completed [55]. Together, replication factor and consistency level allow for tunable consistency supporting Cassandra’s prioritization of availability and partition tolerance over the “all or nothing” approach of strict consistency [55]. It is important to note that any Cassandra node (i.e., coordinator node) or client connected to a coordinator node can coordinate a read or write operation; the coordinator node determines which Cassandra node or nodes own the data (i.e., replicas) and forwards the read or write request accordingly [55].

##### Mechanisms

Anti-entropy, gossip, etc. are some of the architectural pillars (i.e., mechanisms) supporting Cassandra’s decentralized distributed operations. A review of all of these mechanisms is beyond the scope of the current discussion. However, it is important to note that some of these mechanisms are considered essential for decentralized edge-based sensor network solutions in general [67]. Below, we will provide a brief introduction to Cassandra’s commit log, hinted handoff, gossip protocol, and snitch mechanisms. Our selection of Cassandra for our edge-based solution’s distributed database was heavily influenced by its use of these mechanisms. For additional information regarding these mechanisms refer to Carpenter and Hewitt [55].

When a write operation occurs, Cassandra immediately writes the data to a commit log (i.e., to disk); the commit log is a mechanism supporting Cassandra’s durability via crash-recovery [55]. A write operation is not considered successful unless it is written to the commit log [55]. If a Cassandra node crashes, the commit log is replayed in order to ensure data are not lost [55]. If a write operation is sent to a coordinator node and the Cassandra node owning the partition corresponding to the data’s partition key is unavailable, Cassandra implements the hinted handoff mechanism [55]. Hints are saved on the coordinator node and are sent via hinted handoff once the replica node or nodes are back online [55]. Cassandra utilizes a gossiping protocol to exchange endpoint state information amongst Cassandra nodes [55]. In addition to the gossip protocol, Cassandra also implements a snitch to gather network topology information; Cassandra uses this information to efficiently route read and write operations by determining the relative proximity of Cassandra nodes [55].

##### DataStax Enterprise

Initially, we planned to replicate data from the middle to upper layer of our architecture by deploying a single Cassandra cluster consisting of two DCs (i.e., a transactional DC and an analytic DC). Our first real-world field deployment (June 2017) consisted of a single Cassandra cluster with a 20 Cassandra node transactional DC and a three Cassandra node analytic DC. Unfortunately, intermittent network connectivity (i.e., wireless backhaul network) between the two DCs resulted in Cassandra nodes being deprecated due to unanswered topological gossip state updates. Ultimately, this resulted in a loss of data replication at the Cluster and DC levels. In order to overcome the real-world network connectivity problems commonly encountered during remote environmental monitoring, we needed a solution allowing for cluster-to-cluster (i.e., middle-to-upper layer) replication that was tolerant of faulty backhaul network connectivity.

For subsequent field deployments, we transitioned from Cassandra (i.e., DataStax Community Edition) to DataStax Enterprise (DSE) (https://www.datastax.com/products/datastax-enterprise accessed on 11 March 2022). DSE is an enterprise-grade version of Cassandra providing commercial confidence and extra capabilities such as automatic management services, advanced security, and advanced functionality. Our primary interest in DSE’s advanced functionality was DSE Advanced Replication (https://docs.datastax.com/en/dse/6.0/dse-admin/datastax_enterprise/advReplication/advRepTOC.html accessed on 11 March 2022). DSE Advanced Replication supports the configurable replication of data from source to destination clusters in a manner tolerate of the intermittent loss of backhaul network connectivity. DSE Advanced Replication allows for the configuration of automatic failover, permits, and priority to manage traffic between clusters [68]. Using DSE Advanced Replication, we transitioned from a single cluster with two DCs to a direct cluster-to-cluster implementation. DSE Advanced Replication could be configured to further extend our infrastructure to support additional one-to-one or many-to-one (i.e., cluster(s)-to-cluster) implementations. Although not open-source, DSE grants customers a limited no-fee license (https://www.datastax.com/legal/datastax-enterprise-terms accessed on 11 March 2022) for non-production purposes, without the right to support.

##### Summary

There are three key takeaways regarding the use of Cassandra (i.e., DSE) as the edge storage solution for our IoT-based sensor network: (1) DSE is ideally suited for time series data due to its sequential (i.e., from disk) read and write operations; (2) mechanisms, such as commit log, gossip, hinted handoff, and snitches, allow DSE to support high availability, fault-tolerant, and geographically distributed implementations; and (3) the shared-nothing architecture of DSE, when coupled with DSE Advanced Replication, enables nearly linear horizonal scalability for our edge storage and computing framework.

### 3.5. Embedded Systems

#### 3.5.1. Background

Embedded systems include, but are not limited to, microcontrollers, embedded computers, system-on-chip, computer-on-module, and system-on-module. Typically, embedded systems are inexpensive, low power, small, and have modest capabilities when compared with desktop or laptop computers. The Raspberry Pi is among the most popular embedded systems. As of December 2018, the Raspberry Pi was the world’s third best-selling general-purpose computer [69]. As a system originally intended to teach children computer science, the Raspberry Pi is inherently easy-to-use and inexpensive [69]. Having developed an immense community of users, Raspberry Pi based industrial and scientific projects are commonplace [70,71,72].

Our initial selection of the Raspberry Pi was influenced by the Raspberry Pi’s vast community of users and its widespread use within the industrial and scientific communities. We began our development of a multi-node edge-based solution in October of 2016. At the time, the Raspberry Pi 3 B (i.e., 1.2 GHz 64-bit quad core processor with 1 GB of RAM) was available. We installed and configured DSE on the Raspberry Pi 3 B. However, the modest resources of the Raspberry Pi 3 B resulted in frequent downtime (e.g., hangs, reboots, etc.).

In order to improve reliability, we offloaded the Raspberry Pi’s DSE workload by replacing the Raspberry Pi 3 B with the Asus Tinker Board. The Tinker is like the Raspberry Pi 3 B, with an additional gigabyte of RAM (i.e., 2 GB of RAM total). We found the additional gigabyte of RAM significantly improved DSE performance and reliability. Although the Tinker performed well as a DSE node, its user community is not as large as the Raspberry Pi’s. We spent a disproportionate amount of time configuring the Tinker due to its relatively limited support (e.g., drivers, examples, etc.).

In June 2019, the Raspberry Pi Foundation announced the release of the Raspberry 4 [73]. The Raspberry Pi 4 is available in three configurations (i.e., with 2 GB, 4 GB, and 8 GB of RAM), offers USB 3.0 support, and Gigabit Ethernet connectivity. We recently bench tested the Raspberry Pi 4 (i.e., with 2 GB of RAM) and confirmed DSE performance and reliability was equivalent to the Tinker’s. However, we have not had an opportunity to test the Raspberry 4 in a field setting. Any future efforts on our part would utilize the Raspberry Pi 4 as our DSE node’s embedded system.

#### 3.5.2. Related Work

Cassandra and DSE were established cloud and on-premises NoSQL solution in 2016 However, to the best of our knowledge no one had attempted to deploy Cassandra or DSE, on an embedded system, as an edge-based storage and computing solution supporting remote environmental monitoring. Nonetheless, there have been several publications since 2016 exploring the idea of utilizing the Raspberry Pi and Cassandra for IoT applications [74,75,76]. In 2017, Richardson [74] explored the feasibility of using the Raspberry Pi to host Cassandra in support of IoT applications. Richardson [74] utilized the Raspberry Pi (i.e., with 1 GB of RAM) and virtual machines (i.e., with 1 GB, 2 GB, and 4 GB of RAM) to assess the feasibility and performance impact of hosting Cassandra on modest platforms; a minimum of 2 GB of RAM was identified by Richardson [74] as critical for “in-situ IoT storage” using Cassandra. Also in 2017, Romero Lopez [76] undertook the ambitious endeavor of creating a three node Raspberry Pi (i.e., with 1 GB of RAM) Cassandra cluster, deployed via Docker (https://www.docker.com/ accessed on 11 March 2022), including Apache Spark (https://spark.apache.org/ accessed on 11 March 2022). Romero Lopez [76] concluded the Raspberry Pi did not have enough memory (i.e., RAM) for Cassandra or Spark and recommend 4 GB and 8 GB of memory for Cassandra and Spark, respectively.

In 2018, Ferencz and Domokos [75] introduced a data acquisition and storage system using Cassandra and the Raspberry Pi as an alternative to existing IoT data acquisition and storage solutions. Although their system architecture represented a practical and flexible approach to IoT acquisition and storage, Ferencz and Domokos [75] did not run Cassandra on the Raspberry Pi. Likewise, Ooi et al. [77] utilized the Raspberry Pi and Cassandra to effectively acquire and store sensor network data (i.e., seismic). However, Cassandra was not run on the Raspberry Pi.

### 3.6. Communicaion Infrastructure

There is considerable interest in novel, inherently IoT compatible, wireless technologies (i.e., low-power wide-area networks) for seismic applications [38]. However, in order to minimize complexity, we utilized commercially available IP-based wired and wireless components to connect our digitizers, edge devices, edge gateways, and edge nodes. The ports used by Cassandra and DSE for cluster communication and the port used by our digitizers are all IP-based; Cassandra and DSE use TCP and the REF TEK 130-01 uses UDP. See Figure 7 for an overview of the communication infrastructure used for the case study presented in the following section.

An important point to consider, when using embedded systems for remote environmental monitoring, is that their onboard Wi-Fi capabilities are typically inadequate for real-world deployments. Typically, remote environmental monitoring requires that embedded systems and other electronics be placed within enclosures located on or near the ground, in which case the quality of wireless connectivity may degrade. Ideally, embedded systems would connect to a Wi-Fi antenna external to the enclosure. Unfortunately, the Raspberry Pi required board-level modification to connect an external Wi-Fi antenna. The Tinker did allow for the connection of an external antenna via a MHF4 connector. However, the onboard Wi-Fi of the Raspberry Pi (i.e., Raspberry Pi 3 B) and the Tinker did not support our desire to utilize 802.11ac standard communication infrastructure.

A USB Wi-Fi dongle (i.e., TP-Link Archer T2UH AC600 (https://www.tp-link.com/us/home-networking/adapter/archer-t2uh/?utm_medium=select-local accessed on 11 March 2022)) was used to circumvent embedded system Wi-Fi limitations. The TP-Link Archer T2UH AC600 allowed for the connection of an external Wi-Fi antenna and utilized the 802.11ac standard. However, the Tinker did not support the use of the TP-Link Archer T2UH AC600. Ultimately, external antenna capable, 802.11ac standard, Wi-Fi connectivity was achieved by connecting the Tinker to a radio (i.e., EnGenius ENS500EXT-AC (https://www.engeniustech.com/engenius-products/enturbo-outdoor-5-ghz-11ac-wave-2-wireless-access-point/ accessed on 11 March 2022)) via its Ethernet port. The Raspberry Pi used the TP-Link Archer T2UH AC600 to achieve external antenna capable, 802.11ac standard, Wi-Fi connectivity.

## 4. Case Study

### 4.1. Background

Our edge storage and computing framework for IoT-based sensor networks was developed over the course of four test and evaluation (T&E) events occurring in May 2017 (Eastland Lakes, TX, USA), June 2017 (Soda Lake Geothermal Field, NV, USA), July 2018 (Baylor Research and Innovation Collaborative, TX, USA), and May 2019 (San Emidio Geothermal Field, NV, USA). Our first deployment to a geothermal field (i.e., Soda Lake Geothermal Field) consisted of 20 seismic stations deployed along a line approximately 575 m in length and our second deployment to a geothermal field (i.e., San Emidio Geothermal Field) consisted of 144 seismic stations (i.e., planned) deployed along a line approximately 2100 m in length. The case study described below is specific to the San Emidio Geothermal Field T&E event that occurred in May 2019.

### 4.2. Edge Storage and Computing Workflow

A brief overview of the responsibilities of the layers (i.e., workloads) of the edge storage and computing framework was provided in the Framework Overview section of this paper. What follows is a detailed description of the actions performed by the sensing, transactional, and analytic workloads. The sensing workload is responsible for the acquisition of raw seismic data (i.e., three-channels at 250 sps) and the storage of this data in an archive (i.e., RTPD archive) maintained by the edge gateway. The RTPD archive is maintained “as is” in order to keep an original copy of the raw seismic data. A file watcher running on the edge gateway is used to monitor the RTPD archive. When a RTPD file closes, the file (i.e., five-minute file) is copied to a preprocessing directory. Every five minutes, files from the preprocessing directory are read, converted, processed, and data for the vertical channel (i.e., a single channel at 50 sps) is saved as a Comma-Separated Value (CSV) file formatted for insertion into the transactional workload DSE cluster (i.e., transactional cluster). The edge gateway connects to the transactional cluster, via a coordinator node, and writes the CSV data into the cluster. The data are then replicated across the transactional cluster according to a user defined replication factor of two. Two copies of the data are saved on the transactional cluster.

DSE Advanced Replication is configured to replicate data from the transactional cluster to the analytic workload DSE cluster (i.e., analytic cluster). If the backhaul network connectivity between the transactional and analytic clusters is down, the transactional cluster maintains the data needing replication until backhaul connectivity is reestablished. With connectivity reestablished, the transactional cluster replicates data to the analytic cluster. The data are then replicated across the analytic cluster according to a user defined replication factor of two. Two copies of the data are saved on the analytic cluster. See Figure 8 for an overview of the San Emidio Geothermal Field T&E event edge-based solution workflow.

Every hour, data from the analytic cluster is queried and extracted for subsequent seismic processing. Query and extract scripts were executed on a Mini PC (i.e., Intel NUC (https://www.intel.com/content/www/us/en/products/boards-kits/nuc.html accessed on 11 March 2022)) that was collocated with the analytic cluster. Extracted data were then automatically copied to a second collocated Intel NUC designated for processing. ANSI processing was carried out by first applying a bandpass filter of 0.01–24 Hz to remove anthropogenic noise and data recorded by each sensor was divided into 20-s windows. Each window was bit- normalized to adjust amplitudes and then cross-correlated with similar data windows from other sensors and results are stacked to produce virtual source gathers. A virtual source gather is a time vs. distance plot in which one station acts as a source, and is placed at the origin, and the other stations serve as receivers and are plotted as a function of the offset distance from the virtual source station. Upon the completion of ANSI processing, virtual source gathers are saved as plots locally on the NUC.

### 4.3. Implementation

#### 4.3.1. Planned

Prior to the San Emidio Geothermal Field T&E event, we leveraged information obtained from our previous three T&E events to identify performance limiters (e.g., ingest rates, network capacity, etc.) that could not be easily overcome without significant upgrades to solution hardware (i.e., communication infrastructure and embedded systems). Likewise, we considered other physical limiters such as internode spacing of seismic stations, overall length of the seismic line, topography, and operational constraints (e.g., vehicle access, weather, etc.). We considered these limiters in tandem with our geoscientific requirements to organize edge devices, edge gateways, and edge nodes, layer-by-layer (i.e., lower, middle, and upper), into an edge-based solution allowing us to acquire, transmit, store, and process seismic data hourly, in a field setting. The edge-based solution was then replicated and scaled up until it totaled 144 seismic stations. Figure 9 illustrates a “headquarters” consisting of four upper layer edge nodes (i.e., analytic cluster) connected wirelessly to a “squadron” consisting of four middle layer edge nodes (i.e., transactional cluster), in turn, connected (i.e., wired or wirelessly) to twelve lower layer edge gateways, in turn, connected (i.e., wired) to 36 edge devices. Each squadron was responsible for acquiring approximately 778 million samples per day. However, only approximately 156 million samples per day were inserted into the transactional cluster (i.e., squadron) and subsequently replicated to the analytic cluster (i.e., headquarters).

#### 4.3.2. Actual

We intended to deploy four squadrons and four headquarters totaling 144 lower layer edge devices (i.e., seismic stations), 48 lower layer edge gateways, 16 middle layer edge nodes, and 16 upper layer edge nodes. Ultimately, we only deployed 142 seismic stations due to broken or missing REF TEK 130-01 components and middle layer and upper layer components for two (i.e., squadron #1 and squadron #2) of the four planned squadrons. Figure 10 shows a lower layer station consisting of an edge device and edge gateway and Figure 11 shows a middle layer edge node. Weather-related delays and unplanned troubleshooting (i.e., digitizers and communication infrastructure) were primarily responsible for our inability to deploy all four squadrons. However, the organization of squadrons and headquarters into groups, operating independent of each other, provided an opportunity to assess the performance and suitability of our edge-based solution regardless of the total number of squadrons deployed.

#### 4.3.3. Communication Infrastructure

During the San Emidio Geothermal Field T&E event, we deployed our edge-solution in three different communication infrastructure configurations (i.e., “wired”, “hybrid”, and “wireless”) corresponding to T&E event test blocks. The three different configurations allowed us to assess system (e.g., distributed database, embedded system, etc.) performance “layer-by-layer” as we transitioned from predominantly wired to predominantly wireless infrastructure. All three configurations utilized wireless (i.e., a point-to-point wireless bridge) for backhaul network cluster-to-cluster (i.e., middle-to-upper layer) replication; likewise, all three configurations utilized wired (i.e., Ethernet cables) for lower layer edge device to edge gateway connectivity. Headquarters edge nodes (i.e., analytic cluster) were collocated and always connected to each other using wired (i.e., Ethernet cable) connections.

The “wired” configuration utilized Ethernet cables to connect lower layer edge gateways to middle layer edge nodes and middle layer edge nodes (i.e., the transactional cluster) to each other. The “hybrid” configuration continued to use Ethernet cables to connect lower layer edge gateways to middle layer edge nodes. However, middle layer edge nodes (i.e., the transactional cluster) were connected to each other wirelessly (i.e., a WDS access point). Lastly, the “wireless” configuration connected middle layer edge nodes (i.e., the transactional cluster) to each other wirelessly (i.e., a WDS access point) and lower layer edge gateways were also connected to middle layer edge nodes wirelessly (i.e., a wireless access point). Table 1 provides an overview of communication infrastructure configuration. “Wired” and “hybrid” test blocks were conducted for squadron #1 and “wired”, “hybrid”, and “wireless” test blocks were conducted for squadron #2.

## 5. Results

The overall effectiveness of the edge storage and computing framework for IoT-based sensor networks can be assessed by considering its performance and suitability. The solution’s performance refers to quantifiable metrics associated with its ability to function as intended and its suitability refers to its ability to operate in its intended environment.

### 5.1. Performance

#### 5.1.1. Architecture and Topology Performance

Assessing the performance of the edge-based solution’s architecture and topology quantitatively is challenging. Our selection of a Tenet principle inspired architecture and the tree/star hub-and-spoke topology was not driven by specific performance requirements; rather, our choice of architecture and topology evolved over the course of our four T&E events. Nonetheless, we believe the architecture and topology of our edge-based solution supports the implementation non-application specific solutions that complement the constraint driven nature of remote environmental monitoring.

#### 5.1.2. Distributed Database Performance

As an established, enterprise-grade, NoSQL solution, DSE and DSE Advanced Replication proved to be an ideal choice as our framework’s distributed database. Once we determined the absolute minimum hardware requirements (i.e., 2 GB of RAM total), DSE and DSE Advanced Replication performed reliably. Although running on hardware well below its recommended minimum requirements, DSE supported our write-heavy application and provided the performance, reliability, and scalability we required. Layer-by-layer performance data will be provided in the following section. 

As a horizontally scalable distributed database (i.e., NoSQL), DSE complemented our architecture and topology by allowing us to empirically determine our squadron configuration based upon hardware constraints and performance requirements. Essentially, we added/removed nodes, branched, and cut to meet the performance requirements necessary for our specific seismic method. DSE performed consistently and reliably as we arranged and rearranged our squadron configuration over the course of four T&E events. 

#### 5.1.3. Layer Performance

##### Lower Layer Performance

Although weather-related delays were primarily responsible, unplanned troubleshooting also impacted our ability to deploy middle layer and upper layer components for the planned four squadrons. Digitizer (i.e., the REF TEK 130-01) problems (i.e., GPS week number rollover [78] and bad backup battery problems) were relatively easy to solve. However, they were difficult to diagnose. Without the ability to remotely configure the REF TEK 130-01 we were forced to visit seismic stations multiple times before the lower layer components were fully functional. The REF TEK 130-01 can be configured remotely using software provided by the vendor. However, we had not configured the REF TEK 130-01 and the Raspberry Pi (i.e., the edge gateway) to allow remote access to REF TEK 130-01.

Once faulty components were replaced (i.e., the GPS antenna and backup batteries) and the REF TEK 130-01s reconfigured, the lower layer performed its sensing workload as expected. No edge gateway hardware (i.e., the Raspberry Pi) or software (e.g., operating system, Python script, etc.) failures were observed. However, there were instances in which faulty lower layer components required troubleshooting or needed to be replaced (e.g., Ethernet cables, Ethernet switches, etc.). 

##### Lower Layer to Middle Layer Performance

For the “wired” configuration approximately 99.7% and 98.9% of the data acquired by the edge devices was inserted, by the edge gateways, into the edge nodes (i.e., the transactional cluster), for squadron #1 and #2, respectively. Approximately 99.2% and 99.9% of “hybrid” configuration data acquired by the edge devices was inserted, by the edge gateways, into the edge nodes (i.e., transactional cluster), for squadron #1 and #2, respectively. Lastly, the “wireless” configuration resulted in approximately 85.0% of the data acquired by the edge devices being inserted, by the edge gateways, into the edge nodes (i.e., transactional cluster) for squadron #2. Table 2 provides an overview of lower layer to middle layer performance metrics.

We transitioned from 802.11n to 802.11ac standard communication infrastructure for the San Emidio Geothermal Field T&E event. Unfortunately, compatibility problems with the edge gateway’s external antenna and USB Wi-Fi dongle prevented us from deploying our wireless configuration as planned. Spare radios (i.e., the EnGenius ENS500EXT-AC), from our two undeployed squadrons, were used to connect edge gateways wirelessly, via their Ethernet port. However, modifications to the edge gateway “wireless” communication infrastructure were performed in a field setting and required troubleshooting that we believe negatively impacted overall “wireless” configuration performance. 

During the San Emidio Geothermal Field T&E event, each edge gateway was responsible for inserting 45,000 samples of seismic data into DSE every five minutes. A minimum rate of 150 sps (i.e., per edge gateway) was required to ingest data into DSE faster than it was created by edge devices. However, data processing overhead, distributed database-related mechanisms, and variations in network capacity could affect ingest rates. The ratio of edge devices, edge gateways, and edge nodes (i.e., 36:12:4) was adjusted, prior to the T&E event, to allow for a least five times (i.e., 750 sps) the required minimum ingest rate.

Ultimately, our use of 802.11ac standard communication infrastructure supported edge gateway ingest rates ranging from approximately 1200 to 1900 sps, depending upon the communication infrastructure configuration. Approximately eight to twelve times the required minimum ingest rate was available during the San Emidio Geothermal Field T&E event. This provided adequate network capacity to support increasing the number of devices per edge gateway, the ingestion of additional device channels (i.e., the horizontal channels), or increasing the sampling rate of data ingested into DSE.

##### Middle Layer to Upper Layer Performance

For squadron #1 and squadron #2, 100% of “wired” test block data inserted into the transactional cluster was replicated to the analytic cluster (i.e., headquarters #1 and headquarters #2), via DSE Advanced Replication. Approximately 99.9% of squadron #1 and 100% of squadron #2 “hybrid” test block data were replicated to their respective analytic cluster (i.e., headquarters #1 and headquarters #2), via DSE Advanced Replication. Lastly, 99.5% of squadron #2 “wireless” test block data were replicated to its analytic cluster (i.e., headquarters #2), via DSE Advanced Replication. The DSE Advanced Replication backlog was monitored during the “wired”, “hybrid”, and “wireless” test blocks; the DSE Advanced Replication backlog never exceeded more than a few 1000 writes (i.e., samples). Table 3 provides an overview of middle layer to upper layer performance metrics.

Note that only 85.0% of squadron #2 “wireless” test block data acquired by edge devices was inserted into its transactional cluster. However, 99.5% of squadron #2 “wireless” test block data were replicated from its transactional cluster to analytic cluster. This represents an anomaly where more edge device data resided within an analytic cluster than its corresponding transactional cluster. Although the exact cause of this anomaly remains unknown, we believe the anomaly is a result of the independent communication links (i.e., point-to-point wireless versus WDS access point) and the different mechanisms (e.g., commit log, hinted handoff, etc.) used by DSE versus DSE Advanced Replication (i.e., change-data-capture).

##### Upper Layer Performance

An automated script (i.e., Python) was used to query and extract seismic data hourly from the analytic clusters. Leveraging the Python Cassandra driver (https://docs.datastax.com/en/developer/python-driver/3.18/ accessed on 11 March 2022), seismic data were queried from the two analytic clusters (i.e., headquarters #1 and headquarters #2) in parallel and CSV files were extracted for subsequent seismic processing. The query and extract scripts were executed on an Intel NUC and the extracted CSV files were then automatically copied to a second Intel NUC designated for seismic processing. The query and extract of one hour’s worth of squadron data (i.e., 36 seismic stations or approximately 6.5 million samples) took approximately ten minutes, at a rate of approximately 11,000 sps. This provided up to 50 min to perform seismic processing before the next one hour’s worth of data was available for query and extract.

#### 5.1.4. Results from Ambient Noise Seismic Interferometry

In May 2019, we carried out Ambient Noise Seismic Interferometry (ANSI) using the 142-element RaPiER array at the San Emidio Geothermal Field. Figure 12 shows virtual source gathers in which body waves were retrieved from ambient seismic noise using all recorded data in near-real-time. The body wave arrival in Figure 12a,b has a velocity of 380 m/s. The arrival in Figure 12a is not seen beyond ~0.6–0.7 km. This limitation was overcome, and the arrival was observed to nearly twice that distance, by using selective processing methods. Selective stacking of time windows that are deemed the “best” by certain quantitative measures could be automated, along with additional performance assessments, in seismic WSN deployments. See Thangraj and Pulliam (2021) for details of seismic processing and analysis of results. Extracting body wave arrivals from ambient noise seismic data is challenging but with the RaPiER array we were able to produce results with high signal-to-noise qualities using just three days of data and thus validate the acquisition parameters chosen for our seismic deployment.

### 5.2. Suitability

From a suitability perspective, we are confident the San Emidio Geothermal Field T&E event represented a real-world remote environmental monitoring use case. It was necessary for us to deploy temporary infrastructure supporting the operation of edge devices, edge gateways, and edge nodes (i.e., edge components). Antenna masts, batteries, enclosures, and solar panels were deployed to support the continuous acquisition, transmission, storage, and processing of data, without the need to service edge components. 

#### 5.2.1. Power

Our power related support infrastructure relied on one 60 Amp-hour battery and a 20 W solar panel for each edge device (i.e., REF TEK 130-01) not collocated (i.e., not sharing a battery) with an edge gateway (i.e., Raspberry Pi), one 60 Amp-hour battery and 20 W solar panel for each edge device and edge gateway pair (i.e., sharing a battery), and two 60 Amp-hour batteries and a 60 W solar panel for each edge node (i.e., Tinker).

We estimated the overall power draw, via bench testing, of an edge device not collocated with an edge gateway at approximately 2 W, an edge device and edge gateway pair at approximately 4 W, and an edge node at approximately 8 W. The variability in power draw is a result of the various components, configurations, and workloads of the edge components. Without considering solar charging, we estimated a minimum of six days’ worth of available power for edge devices not collocated with an edge gateway, three days’ worth of power for edge device and edge gateway pairs, and three days’ worth of power for edge nodes. 

Our power estimates proved to be accurate. During the San Emidio Geothermal Field T&E event we experienced more than three days of continuous cloud coverage that limited solar charging. We observed low voltage conditions that triggered solar charge controller power cycling (i.e., load off) until battery voltage was restored. This resulted in the temporary loss of a few edge components, typically in the early morning, until power was restored later that morning.

#### 5.2.2. Environmental

Over the course of our four T&E events, we have had ample opportunity to assess the environmental suitability of the edge-based solution. We deployed the equipment in temperatures that ranged from approximately 1 °C to 48 °C and in weather that included dry, dusty, rainy, sleeting, and windy conditions.

Provided they are protected from moisture, commercially available components can usually operate across a wide range of temperatures and environmental conditions. The EnGenius ENS500EXT-AC and the Raspberry Pi’s operating temperatures are −20 °C to 60 °C and −25 °C to 80 °C, respectively. However, we did experience temperature-related failures (i.e., overheating) when deploying other commercially available components (i.e., home or lab use) such as DC to DC converters, ethernet switches, etc. during our first two T&E events. Ultimately, we transitioned to industrial-use components with operating temperatures more closely aligned with the Raspberry Pi’s operating temperature. We did not experience any problems related to environmental conditions during our last two T&E events.

## 6. Discussion

The edge storage and computing framework presented utilizes easy-to-use, inexpensive, and well-established commercially available components and a popular distributed database to orchestrate an edge-based solution for IoT-based sensor networks. Moreover, our use of an architecture inspired by the Tenet principle and the tree/start hub-and-spoke topology supports highly configurable, general-purpose solutions that meet the demands of constraint-driven applications such as remote environmental monitoring. Metrics acquired during the San Emidio Geothermal Field T&E event indicate that the solution supported the in situ acquisition, transmission, storage, and processing of seismic data. As a result, we believe the use of embedded systems (i.e., the Raspberry Pi and Tinker), Mini PCs (i.e., the Intel NUC), DSE, and DSE Advanced Replication to implement an edge-based solution that reduces the cost and risk associated with seismic methods is tenable. We believe our edge-based solution offers two distinct advantages over frameworks that rely on a centralized model (i.e., cloud) or utilize vertically scalable databases (i.e., SQL): (1) it is better suited for remote environmental monitoring (i.e., constraint-drive applications) and (2) it scales from a performance (i.e., node count) and from a geographical perspective.

As geoscientists, our need to design, develop, and deploy an edge-based solution to acquire and process seismic data in a field setting was strongly influenced by our method of seismic monitoring. We planned to utilize a cost effective and non-invasive exploration method using ambient (i.e., passive) seismic noise to characterize the subsurface. One of the primary challenges of using passive (i.e., anthropogenic or natural) seismic noise sources is not knowing the characteristics of the noise sources in advance. As a result, it is impossible to know when you have acquired enough data to successfully characterize the subsurface without first processing and analyzing the data.

The edge-based solution described here minimizes the cost and mitigates the risk typically associated with passive methods of seismic exploration. With data in hand, in a field setting, myriad possibilities are available to leverage conventional and bleeding edge methods to generate higher quality data products. For a summary of framework features see Table 4.

Although we successfully demonstrated an effective edge storage and computing framework for IoT-based sensor networks, this edge-based solution is not without limitations. For instance, the deployment of conventional sensor networks (i.e., wired or wireless) is often logistically challenging. The effort required to prepare, mobilize, and deploy 142 seismic stations for the San Emidio Geothermal Field T&E event was significant. The cables, digitizers, enclosures, power systems, and sensors required for seismic monitoring are expensive, sizeable, and often require specialized knowledge to configure, deploy, and maintain. Commercially available embedded systems and communication infrastructure are relatively easy-to-use and inexpensive. However, they add to the overall logistical burden of sensor network deployments. The value of edge storage and computing must be weighed carefully against its logistical impact.

Commercially available digitizers used in geoscience applications, such as the REF TEK 130-01, are typically very reliable and do not require a lot of supervision. Although Cassandra and DSE support high availability and fault tolerant implementations, our use of embedded systems to host DSE at the edge represents a novel implementation that required continuous oversight during the San Emidio Geothermal Field T&E event. DSE OpsCenter (https://www.datastax.com/products/datastax-enterprise/dse-opscenter accessed on 11 March 2022) is an enterprise-grade management and monitoring solution for DSE clusters. However, the OpsCenter client is not available for embedded systems (i.e., ARM architecture processors). In order to monitor our edge-based solution, we used Ansible (https://www.ansible.com/ accessed on 11 March 2022) and custom Python code to log performance metrics. However, our performance monitoring did not include an overview dashboard. Instead, we were forced to manually review log files throughout our T&E event. We recommend using an overview dashboard, such as OpsCenter, to monitor the overall status of the edge-based solution.

The San Emidio Geothermal Field T&E event provided an opportunity to assess, at hourly intervals, our edge-based solution and the quality of our deployment strategy (i.e., process) in a geoscience application. Unfortunately, an abundance of data can often result in “analysis paralysis” that stifles the decision-making process. When confronted with large data rates, we learned that we needed a strategy that automated the assessment edge-based products and process.

Typically, geoscience-related products are generated using “human-in-the loop” systems that exploit the domain expertise of geoscientists. We believe the automatic generation of geoscience-related products, using an edge-based solution, require specialized methods to objectively assess the overall quality of the products. These specialized methods (e.g., artificial intelligence, statistical, etc.) are necessary to support the relatively rapid operational tempo afforded by an edge-based solution. Likewise, we believe an automated edge-based (i.e., decentralized) version of a seismic quality control program, similar to the program put forth by Ringler et al. [79], allowing for the timely identification and communication of data quality problems would benefit the edge-based solution.

## 7. Conclusions

In this paper we presented an edge storage and computing framework that leverages commercially available communication infrastructure, digitizers, and embedded systems and demonstrated that they can provide valuable capabilities for ambient noise seismic interferometry. The framework is organized in a tiered architecture, arranged in a hub-and-spoke topology, and hosts a popular distributed database to support the acquisition, transmission, storage, and processing of IoT-based sensor network data. We provided details regarding the selection of the architecture, distributed database, embedded systems, and topology used to implement the solution. Lastly, a real-world (i.e., geoscience) case study was presented that leveraged the edge storage and computing framework to acquire, transmit, store, and process millions of samples of seismic data per hour. More than 99% of the data acquired by edge devices (i.e., seismic stations) was stored, queried, and extracted from edge nodes for subsequent seismic processing, in a field setting.

The release of the Raspberry Pi 4 further complements the architecture and topology of our edge-based solution. The availability of three inherently compatible Raspberry Pi 4 versions, with differing capabilities (i.e., RAM), eliminates the need to use different types of embedded systems and Mini PCs for different layers (i.e., lower, middle, and upper), thus simplifying the overall effort required to design, develop, and deploy an edge-based solution.

More importantly, the 4 GB and 8 GB versions of the Raspberry Pi 4 provide an easy-to-use, inexpensive, and well-established embedded system to support an edge-based implementation of Apache Spark. Spark, accepted as an ASF top level project in February 2014, is the most actively developed, open source, unified computing engine for the parallel processing of data on a computer cluster [80]. Spark manages and coordinates the execution of tasks across a cluster of computers [80]. Leveraging the pooled resources of a computer cluster, often in conjunction with a distributed datastore, Spark can process data that a single computer typically cannot [80].

Our edge-based solution is already capable of implementing Spark. DSE provides additional out-of-the-box capabilities, via DSE Analytics (https://docs.datastax.com/en/dse/6.0/dse-dev/datastax_enterprise/analytics/analyticsTOC.html accessed on 11 March 2022), that include Spark integration. Using the Raspberry Pi’s quad-core processor and 4 GB or 8 GB of RAM, upper layer edge nodes could be configured for an analytic workload that leverages the DSE cluster to support Spark (i.e., distributed processing). We have used the Raspberry Pi 4 (i.e., with 4 GB of RAM) to host a four-node, i.e., Spark-enabled, analytic workload DSE cluster and have performed a series of bench tests to assess the feasibility of edge-based distributed processing. We believe the utilization of multiple Raspberry Pi 4s to host a Spark-enabled DSE cluster is feasible and warrants further investigation.

## Figures and Tables

**Figure 1 sensors-22-03615-f001:**
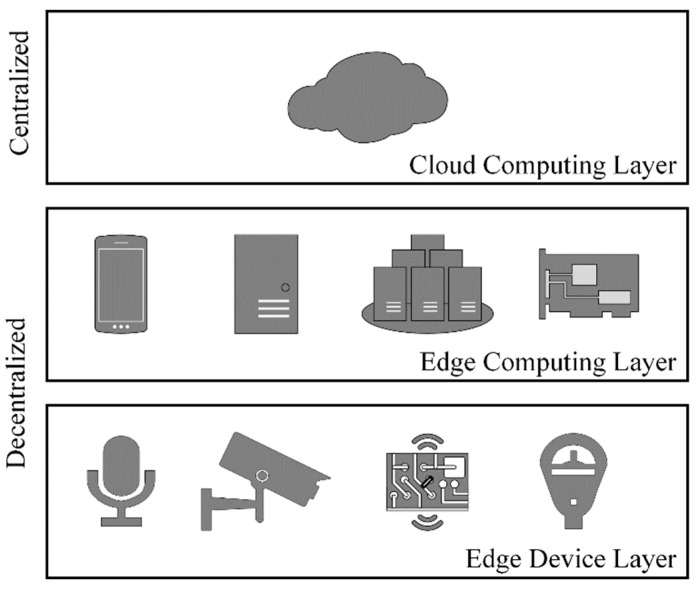
Generic edge computing architecture.

**Figure 2 sensors-22-03615-f002:**
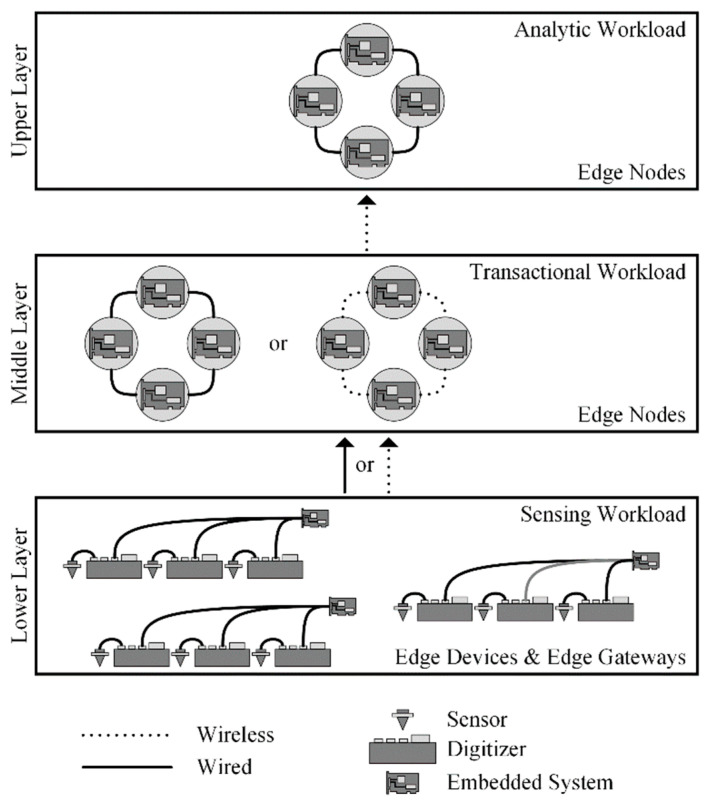
Proposed edge computing architecture.

**Figure 3 sensors-22-03615-f003:**
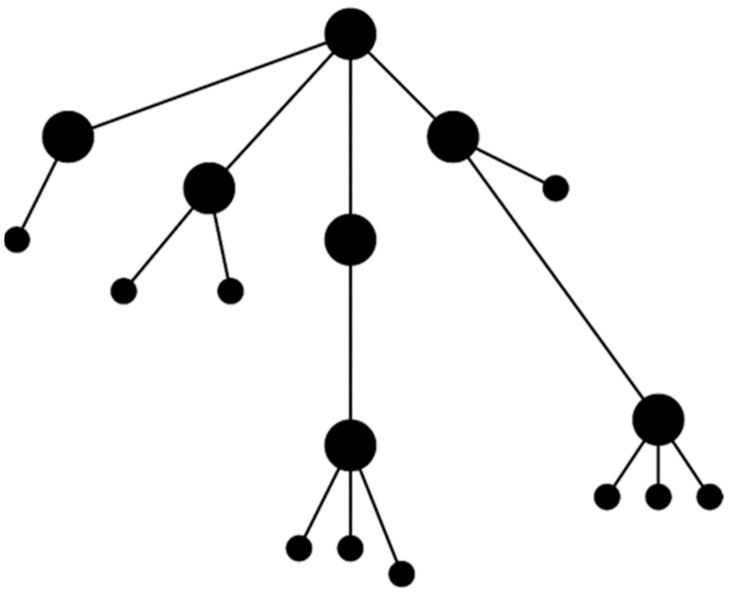
An example of a tree/star network.

**Figure 4 sensors-22-03615-f004:**
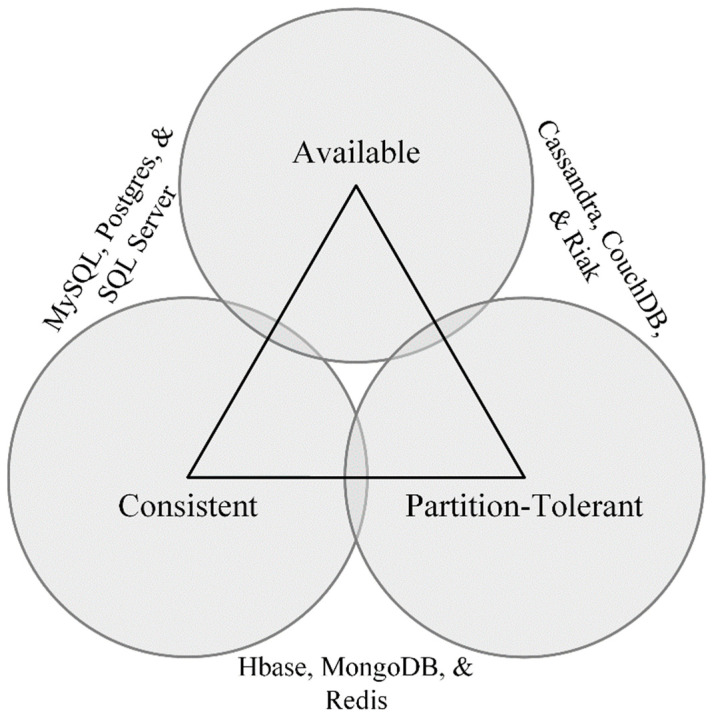
CAP Theorem with examples of datastores positioned along CAP continuum.

**Figure 5 sensors-22-03615-f005:**
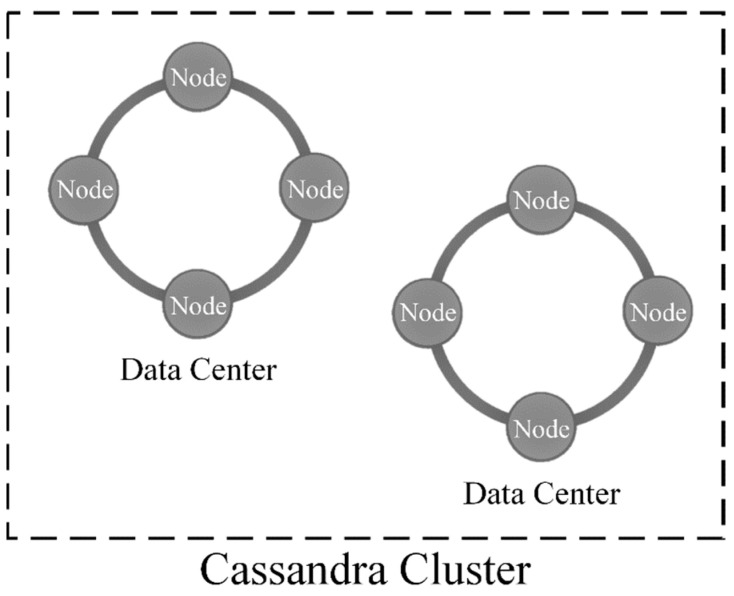
Cassandra cluster, with two DCs, and four Cassandra nodes each.

**Figure 6 sensors-22-03615-f006:**
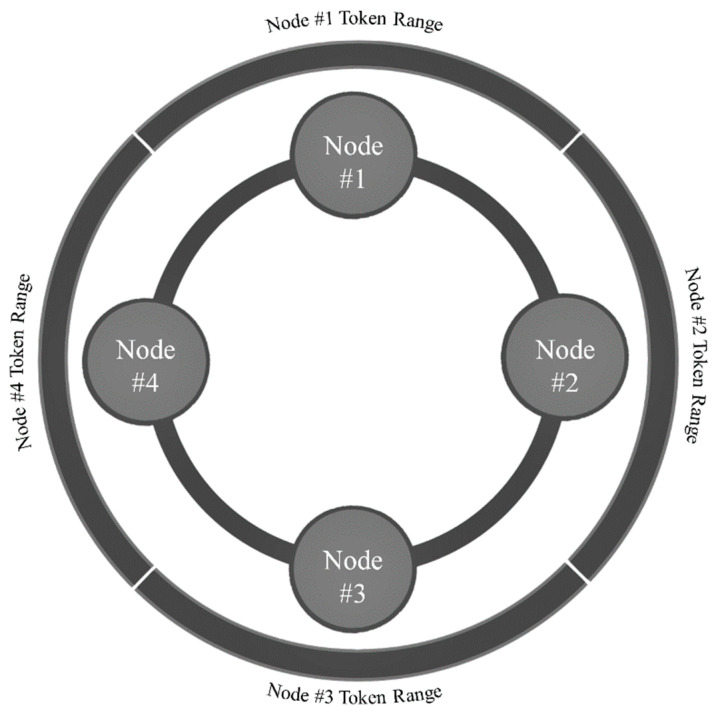
Cassandra cluster and ring shown with four Cassandra nodes and their respective token ranges.

**Figure 7 sensors-22-03615-f007:**
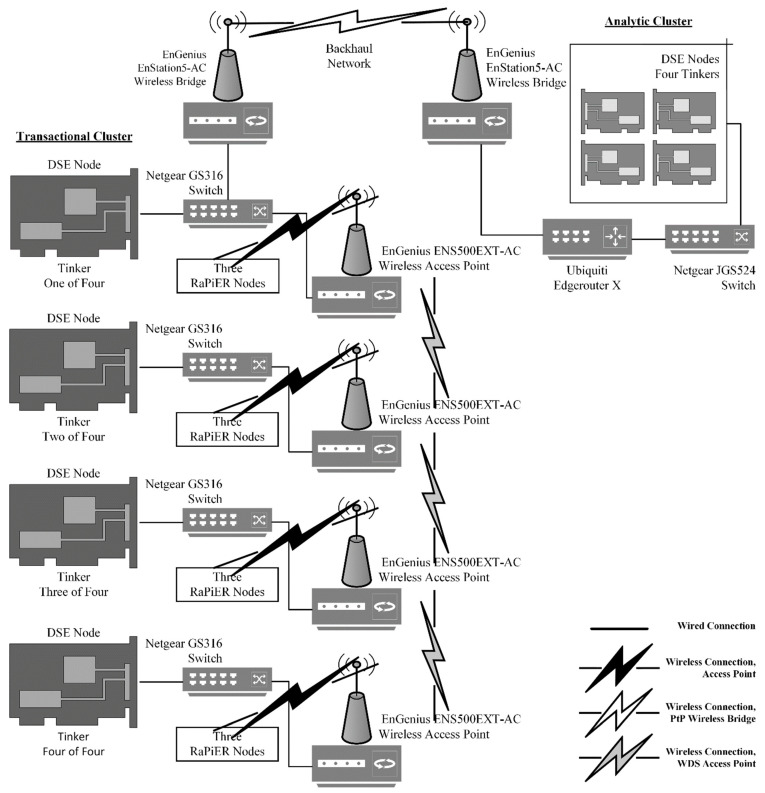
Communication infrastructure “wireless” configuration San Emidio Geothermal Field T&E event case study.

**Figure 8 sensors-22-03615-f008:**
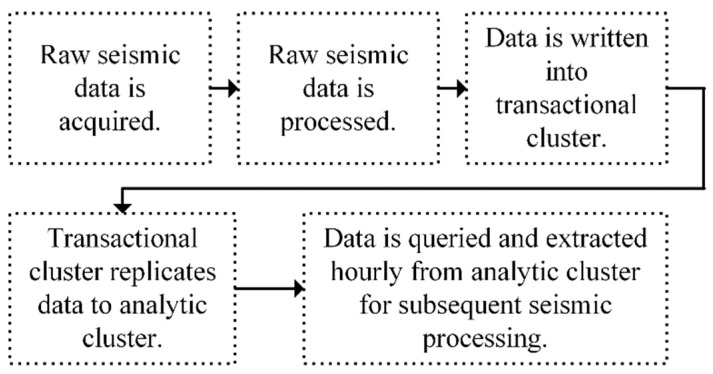
San Emidio Geothermal Field T&E event edge-based solution workflow.

**Figure 9 sensors-22-03615-f009:**
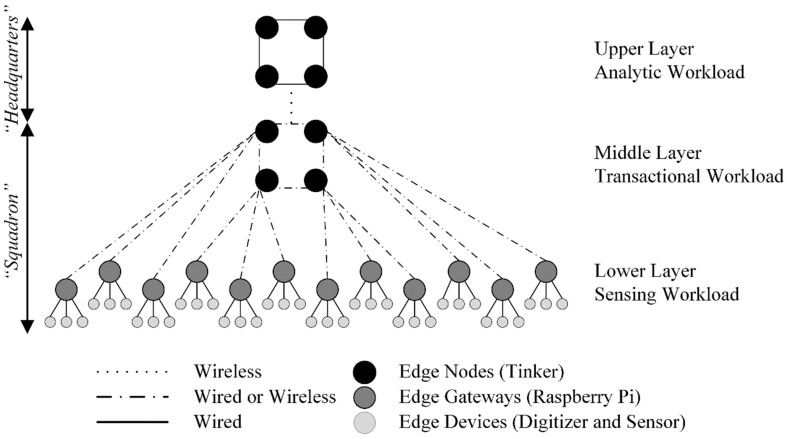
San Emidio Geothermal Field T&E event single squadron and headquarters layout pair.

**Figure 10 sensors-22-03615-f010:**
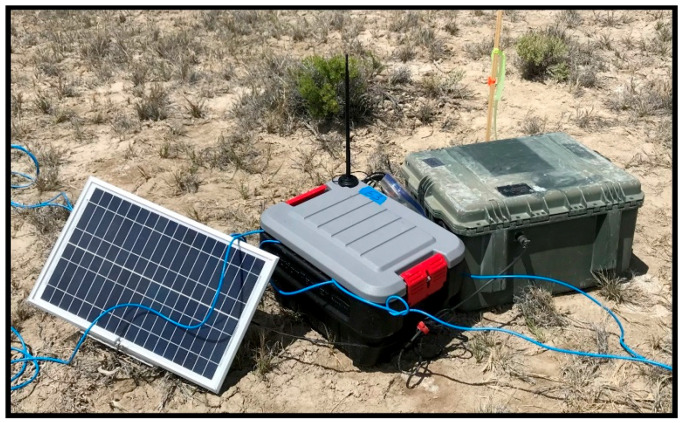
San Emidio Geothermal Field T&E event lower layer edge device and edge gateway.

**Figure 11 sensors-22-03615-f011:**
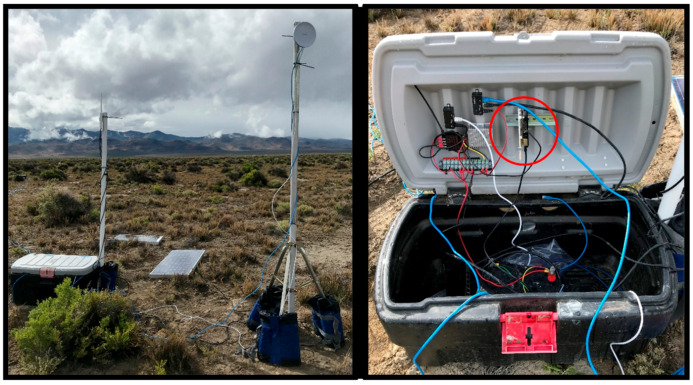
San Emidio Geothermal Field T&E event exterior view of middle layer edge node (**left**) and interior view edge node (**right**). The Asus Tinker is shown in the red circle (interior view) with power-related components to its left (e.g., DC-to-DC, terminal bus, etc.) and a battery and network switch located within the enclosure.

**Figure 12 sensors-22-03615-f012:**
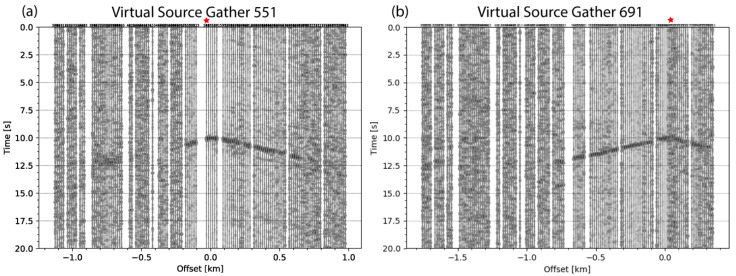
(**a**) Virtual source gather 551 is obtained by cross-correlating data from station 551 with all other stations’ data. (**b**) Virtual source gather 691 is obtained by cross-correlating data from station 691 with all other stations’ data.

**Table 1 sensors-22-03615-t001:** Communication Infrastructure Configuration.

Communication Infrastructure Configuration	Lower Layer to Lower Layer	Lower Layer to Middle Layer	Middle Layer to Middle Layer	Middle Layer to Upper Layer
	(See Figure 9 for Layer Details)			
**“Wired”**	Ethernet Cable	Ethernet Cable	Ethernet Cable	Point-to-Point Wireless Bridge
**“Hybrid”**	Ethernet Cable	Ethernet Cable	Wireless Distribution System Access Point	Point-to-Point Wireless Bridge
**“Wireless”**	Ethernet Cable	Wireless Access Point	Wireless Distribution System Access Point	Point-to-Point Wireless Bridge

**Table 2 sensors-22-03615-t002:** Summary of lower layer to middle layer performance metrics.

Layer	Communication Infrastructure Configuration	Squadron #	% Data Inserted into DSE	Approximate Data Ingest Rate (Writes per Second)
**Lower Layer** **to Middle Layer** (See Figure 9 for Layer Details)	“Wired”	Squadron #1	99.7	1900
		Squadron #2	98.9	
	“Hybrid”	Squadron #1	99.2	1550
		Squadron #2	99.9	
	“Wireless”	Squadron #1	--	1200
		Squadron #2	85.0	

**Table 3 sensors-22-03615-t003:** Summary of middle layer to upper layer performance metrics.

Layer	Communication Infrastructure Configuration	Squadron #	% Data Inserted into DSE
**Middle Layer to** **Upper Layer** (See Figure 9 for Layer Details)	Wired	Squadron #1	100.0
		Squadron #2	100.0
	Hybrid	Squadron #1	99.9
		Squadron #2	100.0
	Wireless	Squadron #1	--
		Squadron #2	99.5

**Table 4 sensors-22-03615-t004:** Summary of edge storage and computing framework features.

Framework Features	
**Architecture**	Utilizes a tiered architecture supporting workloads of varying complexity.
**Topology**	Utilizes a hub-and-spoke topology supporting the addition and/or redistribution of edge nodes to overcome common edge-based performance limiters.
**Distributed Database**	Uses a datastore based upon an open-source solution that: is ideally suited for time series data, supports high availability, fault-tolerant, and geographically distributed implementations, and offers nearly linear horizontal scalability.
**Embedded Systems**	Uses easy-to-use, inexpensive, and well-established commercially available components.
**Communication** **Infrastructure**	Uses commercially available IP-based wired and wireless components.

## Data Availability

Instrumentation and data archiving were provided by the Incorporated Research Institutions for Seismology through their PASSCAL and Data Management Centers, respectively. Both centers provided technical support. All data and associated metadata from our T&E events at Eastland Lakes, Soda Lake Geothermal Field, the Baylor Research and Industrial Collaborative, and San Emidio Geothermal Field have been archived at the Data Management Center (DMC) operated by the Incorporated Research Institutions for Seismology (IRIS). The open-source algorithms and codes used to derive processed and interpreted seismic results, including detailed descriptions of data processing methodology, are available with MSNoise distribution: www.msnoise.org.

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
