# Peer review of "The Edge of Exploration: An Edge Storage and Computing Framework for Ambient Noise Seismic Interferometry Using Internet of Things Based Sensor Networks"

_sensors, 2022, doi:10.3390/s22103615_

Round 1

Reviewer 1 Report

Pls refer to the attached pdf file

Author Response

Please see the attachment.  Response line notations are based on line count within the updated manuscript.

Reviewer 2 Report

In this paper, the authors present an edge storage and computing framework leveraging commercially available components organized in a tiered  architecture and arranged in a hub-and-spoke topology. A real-world case study shows the application value of edge-based technical solutions in the field of earthquake monitoring. Through a detailed review of thispaper, there are several small issues that need to be further improved:
1. Although this work is to provide a technical solution for the engineering field, it would greatly enhance the academic value of this paper if the authors could illustrate some theoretical contributions of the proposed solution.
2. This paper seems to provide background and related work introduction for each section separately, such as Section 3.2, Section 3.3, Section 3.3, Section 3.4, Section 3.5, etc. Can the authors discuss these through a related work section?

Author Response

(The authors gave the same response as above.)

Reviewer 3 Report

The authors presented the utilization of the IoT-based sensor network for ambient noise seismic interferometry. In the introduction, the authors admit to using in their solution the common components, like Raspberry Pi and Tinker Board, which is in my opinion very big advantage. This proves that IoT, embedded systems, as well as other IT can be used in almost all fields of science. 

Authors very precisely described their framework, architecture of the system, used hardware and software components. The only disadvantage is a lack of the separated presentation of similar systems and the clearly stated discussion on using different components.

Experimental results were obtained in real-world application. Discussion and conclusions are exhausting and well elaborated.

Author Response

(The authors gave the same response as above.)

Round 2

Reviewer 1 Report

Most of my concerns have been addressed except point 2 regarding the contributions: I suggest contributions can be re-written to reflect the uniqueness of the proposed system authors claimed in the response:  "1) our framework does not require a connection to the internet for continuous monitoring and 2) our framework can scale (i.e., beyond “…thirteen nodes located a few meters apart and six nodes located approximately 15 meters apart…"
